# The Repurposing of FDA-Approved Drugs as FtsZ Inhibitors against *Mycobacterium tuberculosis*: An In Silico and In Vitro Study

**DOI:** 10.3390/microorganisms12081505

**Published:** 2024-07-23

**Authors:** Andrea Michel Tovar-Nieto, Luis Enrique Flores-Padilla, Bruno Rivas-Santiago, Juan Valentin Trujillo-Paez, Edgar Eduardo Lara-Ramirez, Yolanda M. Jacobo-Delgado, Juan Ernesto López-Ramos, Adrián Rodríguez-Carlos

**Affiliations:** 1Medical Research Unit—Zacatecas, Mexican Institute for Social Security—IMSS, Interior of Alameda 45, Colonia Centro, Zacatecas 98000, Mexico; micheltn25@gmail.com (A.M.T.-N.); rondo_vm@yahoo.com (B.R.-S.); yolandajacobo17@gmail.com (Y.M.J.-D.); 2Centro de Estudios Científicos y Tecnológicos 18 Zacatecas, Instituto Politécnico Nacional, Zacatecas 98160, Mexico; fluis7604@gmail.com (L.E.F.-P.); taneiro87@hotmail.com (J.V.T.-P.); 3Pharmaceutical Biotechnology Laboratory, Genomic Biotechnology Center, Polytechnic Institute National, Reynosa 88710, Mexico; elarar@ipn.mx

**Keywords:** FtsZ inhibitor, *Mycobacterium tuberculosis*, FDA-approved drugs, MDR tuberculosis

## Abstract

*Mycobacterium tuberculosis* (Mtb), the causative pathogen of tuberculosis, remains one of the leading causes of death from a single infectious agent. Furthermore, the growing evolution to multi-drug-resistant (MDR) strains requires de novo identification of drug targets for evaluating candidates or repurposing drugs. Hence, targeting FtsZ, an essential cell division protein, is a promising target. Methods: Using an in silico pharmacological repositioning strategy, four FDA-based drugs that bind to the catalytic site FtsZ were selected. The Alamar Blue colorimetric assay was used to assess antimicrobial activity and the effect of drugs on Mtb growth through growth curves. Bacterial load was determined with an in vitro infection model using colony-forming units (CFU)/mL, and cytotoxicity on human monocyte-derived macrophages (MDMhs) was assessed by flow cytometry. Results: Paroxetine and nebivolol exhibited antimycobacterial activity against both reference TB and MDR strains at a concentration of 25 µg/mL. Furthermore, both paroxetine and nebivolol demonstrated a significant reduction (*p* < 0.05) in viable bacteria compared to the untreated group in the in vitro infection model. Conclusions: Collectively, our findings demonstrate that the use of paroxetine and nebivolol is a promising strategy to help in the control of tuberculosis infection.

## 1. Introduction

Tuberculosis (TB) is a disease caused by the *Mycobacterium tuberculosis* complex bacilli and is one of the oldest diseases known and a major cause of death worldwide from an infectious disease. In 2023, the World Health Organization (WHO) estimated around 10.6 million people would develop a TB infection [1]. While enhanced chemotherapy and diagnosis of tuberculosis have lowered global TB incidence rates by 2% since 2015, the reality is that we are far from achieving the objectives outlined by the End TB Strategy [1,2]. The growing development of drug-resistant tuberculosis (DR-TB) and multi-drug-resistant tuberculosis (MDR) are the most urgent and difficult challenges facing global TB control. Globally, there existed 410,000 cases of DR-TB in 2022, and it is estimated that only 2 in 5 of those cases were enrolled in an adequate treatment program [1]. Therefore, there is an urgent need to develop new compounds, reuse already available drugs, or optimize the doses of first-line anti-tuberculosis drugs.

Given the growing need for new anti-tuberculosis drugs, pharmaceutical repurposing is presented as an alternative [3]. Therefore, many of the disadvantages are reduced, such as the slow and expensive process of traditional de novo drug discovery. Some clinical trials based on drug repurposing have emerged as promising candidates for treating tuberculosis [4]. For example, linezolid, a member of the oxazolidinone family of antibiotics, has been shown to improve patients with MDR tuberculosis [5]. Similarly, clofazimine, used in the treatment of leprosy, has also shown its effectiveness in cases of DR-TB [6]. Therefore, this strategy has proven to be a promising tool to counteract the challenges in antibiotic drug development and the increasing development of DR-TB.

Recently, pharmacological targets based on the indispensability of bacterial proliferation and its prevalence across bacterial species have been recognized as potential targets for antibacterial agents. FtsZ (filamentation temperature-sensitive protein Z) is an essential cell division protein in bacteria encoded by the FtsZ gene. FtsZ’s role in cell division is analogous to actin’s in eukaryotic cell division [7]. The evidence has demonstrated that FtsZ polymerizes in a GTP-dependent manner into filaments, which assemble into a highly dynamic structure known as the Z-ring on the inner membrane at the mid-cell. This process ends with the recruitment of the other cell division proteins and the contraction of the Z-ring, resulting in septation (divisome). Thus, the inactivation of FtsZ results in defective Mtb cell division [8]. Consequently, FtsZ is a promising target for the discovery of new antimicrobial drugs.

Many compounds currently studied in preclinical assays exhibited antibacterial activities targeting FtsZ. For instance, the use of SRI-3072 [9], totarol [10], zantrin [11], berberine, and its derivatives showed anti-Mtb activity by inhibiting the GTPase activity of FtsZ [12]. Similarly, the compounds TB-E12 and 297-F showed FtsZ inhibition activity on Mtb [13]. However, despite this promising potential, the development of FtsZ inhibitors as an antibacterial is still in the early stages [14], and additional research is required to optimize their effectiveness and selectivity. In the present study, we explored the available drug repositories by virtual screening to identify FDA-approved drugs as FtsZ inhibitors against *Mycobacterium tuberculosis*.

## 2. Materials and Methods

### 2.1. Molecular Docking

The crystallized structure of the Ftsz protein was obtained from the Protein Data Bank (PDB) database with identification code 6YM1 that corresponds to the protein of *Mycobacterium tuberculosis* in complexes with guanosine diphosphate (GDP) [15]. The crystal structure was prepared in PyMOL software (the PyMOL Molecular Graphics System, version 2.5.5, Schrödinger, LLC., New York, NY, USA) to remove hydrogen atoms and water molecules. Two files were generated that correspond to the receptor (FtsZ) and the natural ligand of the receptor (GDP).

A conformational space was defined to perform the docking with dimensions of 10 × 18 × 16 centered at x = 50.153, y = −39.833, and z = 19.749. Therefore, a total of 1947 FDA-approved structures were obtained from the Zinc20 database (accessed on 16 May 2023 [16]). Smiles files (Simplified Molecular Input Line Entry Specification) were obtained and the “obabel” command was used to optimize the geometry of the molecules and minimize their energy, obtaining the PDBQT formation from the mol2 files. Then, docking was performed using Autodock Vina version 1.2.0 (Vina). Finally, the vina score of each FDA-approved structure was compared with the energy obtained from the natural ligand (GDP), which was −8.5 kCal/mol.

### 2.2. Cell Culture

The Mtb H37Rv strain (ATCC 27294) and MDR aisled strain were maintained in complete Middlebrook 7H9 medium (Becton Dickinson, Franklin Lakes, NJ, USA) supplemented with 0.2% glycerol (*v/v*) (Fisher Scientific, Hampton, NH, USA), 0.5% Tween80 (*v/v*) (Sigma-Aldrich, St. Louis, MO, USA), and 10% (*v/v*) oleic acid, albumin, dextrose, and catalase (OADC enrichment medium; BBL, Becton Dickinson, Franklin Lakes, NJ, USA) at 37 °C with agitation at 100 rpm. The strain was kept in culture until reaching the logarithmic phase, determined by spectrophotometry.

Macrophages derived from human monocytes (MDMhs) were obtained according to the Declaration of Helsinki and approved by the National Committee of Ethics and the National Commission of Scientific Research of the Mexican Institute of Social Security (IMSS). Briefly, the isolation and differentiation to macrophages procedures were carried out according to previous reports. Peripheral blood mononuclear cells (PBMCs) were isolated using Lymphoprep (StemCell Technologies, Vancouver, BC, Canada). PBMCs (2.5 × 10^6^) were cultured in RPMI medium using 24-well ultra-low adhesion plates (Costar, Corning, NY, USA) [17]. After 2 h, non-adherent cells were removed and subsequently differentiated for 7 days with RPMI medium (Corning Incorporated, Corning, NY, USA) supplemented with 10% decomplemented pooled human serum (Biowest, Riverside, MO, USA).

### 2.3. Microplate Alamar Blue Assay (MABA)

The antimicrobial activity of sitagliptin, atovaquone, paroxetine, and nebivolol was tested by microdilution assay using a resazurin-based oxidation–reduction dye, Alamar Blue, described previously [18]. Briefly, the bacteria were diluted at 1:20 (6.6 *×* 10^6^ CFU/mL) in Middlebrook 7H9 medium, and 100 µL was added to a 96-well microplate. Then, serial 1:2 dilutions were performed with selected drugs, and after 5 days at 37 °C, the Minimal Inhibitory Concentration (MIC) was determined. Next, 44 μM resazurin (Acros Organics 418900050, Geel, Belgium) was added, the plate was incubated for an additional 24 h, and the growth of the bacilli was determined by observing a visual color change. The Alamar Blue dye indicates bacterial growth and metabolism by changing from blue (oxidized, non-fluorescent) to a reduced form, pink (reduced, fluorescent). The conventional antibiotics, rifampicin and streptomycin, were used as growth inhibition controls. The culture medium was used as a blank and bacteria without treatment as a bacteria control. The MIC was defined as the lowest drug concentration which presented a color change from blue to pink.

### 2.4. Evaluation of Growth Inhibition of Mtb

To evaluate the inhibitor activity of the candidate molecules over Mtb growth, H37Rv and MDR strains were cultured in 10 mL of 7H9 medium (Becton Dickinson, Franklin Lakes, NJ, USA) supplemented with 10% oleic acid, albumin, dextrose, and catalase (OADC enrichment medium; BBL, Becton Dickinson, Franklin Lakes, NJ, USA), along with 0.5% Tween 80 and 0.2% glycerol, in 25 cm^2^ culture flasks (Corning Incorporated, Corning, NY, USA), as described previously [19]. The cultures were incubated in the presence and absence of the compounds, the drugs paroxetine and nebivolol were added at a concentration of 25 µg/mL, and the optical density (OD) at 600 nm was measured every 3 days over 21 days. Rifampicin and streptomycin were used as controls for growth inhibition.

### 2.5. Cytotoxicity Assay

The cytotoxicity of the compounds that showed antimicrobial activity in the MABA assays was evaluated in MDMhs. The cells were seeded in 24-well plates with low adhesion with 1 mL of their respective growth media. The plates were incubated to allow for cell adhesion. Subsequently, the supernatants were removed, and new media prepared with different stimuli were added, including the molecules of interest. The cells were incubated at 37 °C with 5% CO_2_ for 48 h. As a positive control, 10% DMSO was used and cells with a complete medium were used as a growth control.

At the end of the stimulation time, the cells were detached from each well individually by washing them with PBS. Then, they were collected with their respective supernatants in tubes for flow cytometry. Subsequently, they were centrifuged for 7 min at 1500 rpm and the new resulting supernatant was discarded. The cells were suspended in 100 μL of the Guava ViaCount Millipore reagent (Burlington, MA, USA), according to the supplier’s instructions, for each condition. The results are expressed as live, apoptotic, and necrotic cell percentages. Finally, the data were analyzed using the FACsDiva software version 6.1.3 on the FACs CANTO II flow cytometer (Becton Dickinson, Franklin Lakes, NJ, USA).

### 2.6. CFUs/mL Assay

MDMhs were infected with Mtb H37Rv or MDR strains at a multiplicity of infection of five (MOI 5:1, bacilli/cell ratio). After 2 h of infection, cells were washed to remove extracellular bacteria. Infected cells were treated with 25 µg/mL and 10 µg/mL paroxetine and nebivolol for 24 h to evaluate CFU/mL in macrophages. Then, the cells were lysed with 0.01% SDS (Boehringer Mannheim, Indianapolis, IN, USA) for 10 min and the reaction was stopped with 20% bovine serum albumin (Biowest, Nuaillé, France) to subsequently make serial dilutions that were plated in triplicate on Middlebrook 7H10 agar (DIFCO, Detroit, MI, USA). Finally, CFU/mL was determined after 14 and 21 days (about 3 weeks) of incubation at 37 °C, with 5% CO_2_.

### 2.7. Statistical Analysis

Statistical analysis was performed using the GraphPad Prism software (GraphPad Software version 6.01 San Diego, CA, USA). Normal distribution was assessed using the Kolmogorov–Smirnov test for each data set, together with a non-parametric multiple comparison test of Kruskal–Wallis to identify differences among the groups. When statistical significance (*p* < 0.05) was found, Dunn’s post-test was performed. *p*-values of < 0.05 were considered statistically significant.

## 3. Results

### 3.1. Molecule Selection

The binding modes of proteins and ligands were studied by measuring the vina scores obtained from the docking tool AutoDock Vina. Twelve molecules exhibited a binding affinity to FtsZ compared with GDP (−8.5 kcal/mol), the compound utilized during the creation of the primary pharmacophore model. From the in silico screening, we identified four drugs with favorable vina scores. The four FDA-approved drugs were sitagliptin, paroxetine, nebivolol, and atovaquone (Table 1). These four molecules were chosen based on their vina score, their number of hydrogen bonds, and their import availability.

The active site of FtsZ was identified on the amino acid residues ASP-184, GLU-136, ARG-140, GLY-19, GLY-18, ANS--22, THR-106, GLY-105, and GLY-107, which have a role in GDP interactions. Table 1 and Figure 1a show the docking interactions with active site amino acid residues. Thus, we observed that nebivolol formed more interactions with FtsZ at active site residues of ASN-22, THR-106, GLY-19, ARG-140, and GLU-136 (Figure 1d), as opposed to sitagliptin with GLU-136 and ARG-140 (Figure 1b), paroxetine with THR-106, GLY-105, GLY-19, and GLU-102 (Figure 1c), and atovaquone with ARG-140 (Figure 1e).

### 3.2. Paroxetine and Nebivolol Show Anti-Mycobacterial Tuberculosis Activity

The in vitro antimicrobial activity of the selected FtsZ inhibitor candidate molecules was tested on the Mtb H37Rv and MDR strains. Table 2 shows the MIC values of the tested molecules compared with streptomycin and rifampicin. We did not observe any antimicrobial activity with sitagliptin and atovaquone at the tested concentrations (150 µg/mL and 10 µg/mL, respectively). However, paroxetine and nebivolol exhibited antibacterial activity against both Mtb strains at 25 μg/mL.

The antimicrobial effects of paroxetine and nebivolol were further evaluated over 21 days at a concentration of 25 µg/mL for both Mtb strains. Our results indicate that paroxetine treatment significantly delays growth up to 4 days (*p* < 0.0001) for H37Rv and up to 7 days (*p* < 0.05) for MDR strains (Figure 2a,c). Similarly, stimulation with nebivolol reduces the logarithmic growth phase starting from day 7 (*p* < 0.0001), which is normally reached by day 14 in the non-stimulated group (Figure 2b,d).These findings demonstrate that nebivolol and paroxetine exhibit a potent antibacterial effect against both sensitive and MDR strains of Mtb.

### 3.3. Paroxetine and Nebivolol Reduce CFU/mL Counts in Macrophages

To investigate the toxic effect of paroxetine and nebivolol on MDMhs, we first evaluated a range of concentrations based on their MIC assay. The results show that treatment with nebivolol and paroxetine at a concentration of 25 µg/mL does not affect cell viability (Figure 3). The use of concentrations greater than 200 µg/mL for paroxetine and 50 µg/mL for nebivolol significantly affects the viability of cells (*p* < 0.05). Thus, we confirmed that concentrations lower than 25 µg/mL were safe for future experiments.

In the present study, the Mtb-infected macrophages were treated with paroxetine and nebivolol to assess their efficacy in controlling bacterial growth. Figure 4a,b demonstrate that the use of paroxetine and nebivolol significantly reduced the total H37Rv bacterial load compared to untreated conditions, but only at higher concentrations. Lower concentrations did not show a significant difference in H37Rv load. Interestingly, both paroxetine and nebivolol facilitated the clearance of the MDR strain in infected macrophages across all concentrations tested (Figure 4c,d).

**Figure 2 microorganisms-12-01505-f002:**
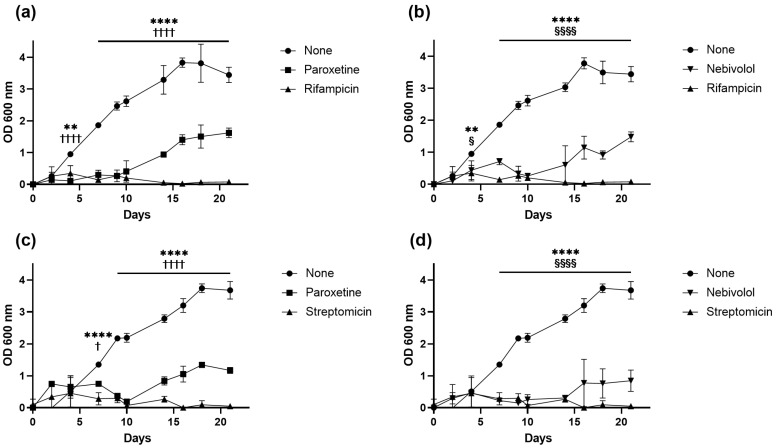
Mycobacterial growth curves in presence and absence of FtsZ inhibitor candidate molecules. Growth curves were monitored by optic density (600 nm) of (**a**,**b**) H37Rv and (**c**,**d**) MDR strains in growth media containing 25 µg/mL nebivolol (**b**,**d**), 25 µg/mL paroxetine (**a**,**c**), streptomycin (0.5 µg/mL), and rifampicin (1 µg/mL), compared with normal growth with none. (** *p*< 0.01, **** *p*< 0.0001 none vs. rifampicin) (§ *p* < 0.05, §§§§ *p*< 0.0001 none vs. nebivolol) (†††† *p*< 0.0001 none vs. paroxetine; † *p* < 0.05 none vs. paroxetine) *n* = 3.

**Figure 3 microorganisms-12-01505-f003:**
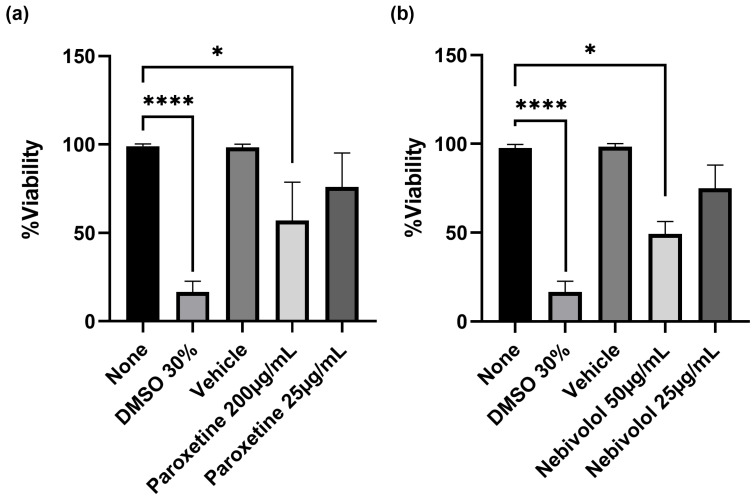
Macrophages’ viability treated with (**a**) paroxetine and (**b**) nebivolol. Cell viability was determined using Guava^®^ ViaCount™ assay. Data are expressed as mean ± standard deviation (SD). Statistics were calculated by Kruskal–Wallis and Dunn’s post hoc tests. In each experimental group, * *p*< 0.05; **** *p* < 0.0001; *n* = 3.

**Figure 4 microorganisms-12-01505-f004:**
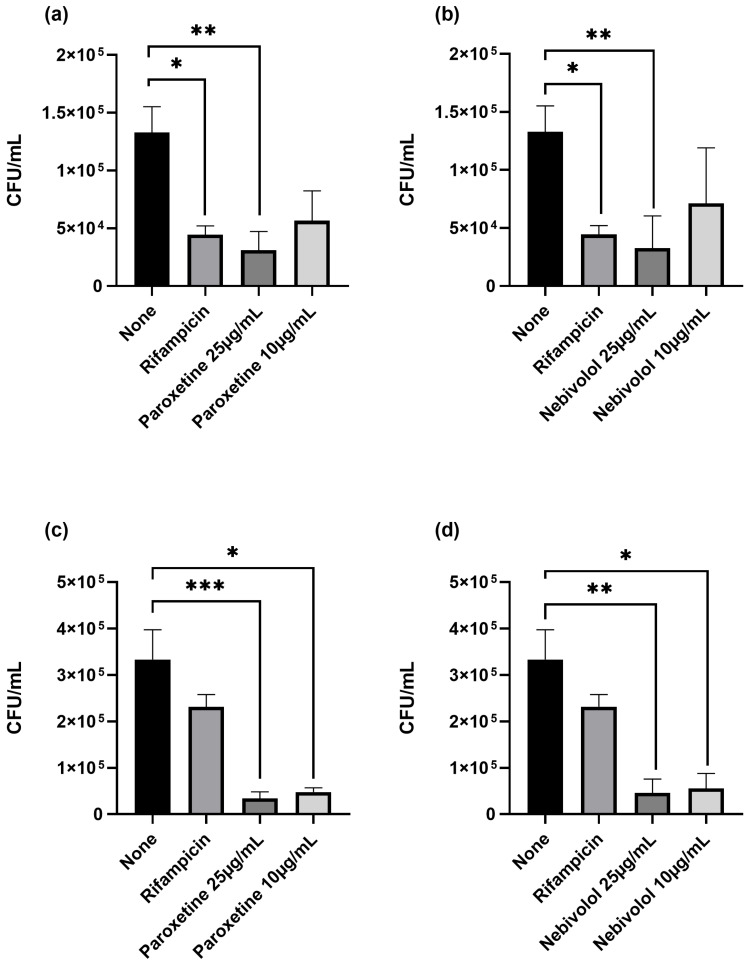
Paroxetine and nebivolol treatment reduce the colony-forming units (CFU)/mL of *Mycobacterium tuberculosis* in macrophages. Macrophages were infected with Mtb at an MOI of 1:5 with H37Rv (**a**,**b**) and MDR (**c**,**d**) strains and treated with 25 µg/mL and 10 µg/mL of paroxetine (**a**,**c**) and nebivolol (**b**,**d**). Rifampicin was used as a control. Cells were lysed and CFU/mL were evaluated. The graphs show the mean ± standard deviation (SD) from six independent experiments by duplicate. Statistics were calculated by Kruskal–Wallis and Dunn’s post hoc tests. * *p* < 0.05; ** *p* < 0.01; *** *p* < 0.001; n = 3.

## 4. Discussion

TB and the increasing prevalence of MDR strains represent a global public health crisis, which is exacerbated by socioeconomic factors, diagnostic challenges, and treatment difficulties, among others. Furthermore, the duration of treatment for DR-TB is approximately three times longer, and the risk of adverse events is increased due to the larger number of pills required. Therefore, the WHO emphasizes the need to adopt innovative therapeutic approaches for DR-TB; some goals are based on shorter duration, lower cost, and fewer pills with greater effectiveness [1]. A focused and rational approach to TB drug discovery involves a comprehensive understanding of the pathogen at the molecular level, specifically the pathways involved in cell division and proliferation.

This knowledge helps in identifying the molecular targets or pathways that are critically involved in a disease state to solve the problem of DR-TB [24]. Among these targets, the FtsZ protein plays a crucial role in the survival and proliferation of Mtb. FtsZ is essential for the polymerization-competent tubulin homolog that localizes to the division site to form a structure known as the Z-ring [25]. This enzyme is pivotal in regulating cell wall biosynthesis and division, making it an attractive candidate for drug development, leading to impaired bacterial reproduction. Inhibiting FtsZ in bacteria has significant consequences due to its critical role in bacterial cell division. This can result in growth arrest or even bacterial death.

Molecular docking has become one of the most widely utilized tools in drug discovery, as it allows researchers to predict how small molecules, or ligands, interact with biological targets such as proteins. This technique is particularly valuable for understanding the significance of ligand binding and the potential inhibition of protein functions. Despite its advantages, molecular docking has limitations and often requires experimental validation to confirm the predicted interactions [17,26]. In this study, the molecular docking of paroxetine and nebivolol over the FtsZ protein demonstrated a relatively better binding dynamics-based protein–ligand interaction with the same amino acids that may be attributed to GTPase activity.

Sitagliptin is an oral medication that acts as a selective inhibitor of dipeptidyl peptidase-4 (DPP-4), also known as CD26. This enzyme plays a crucial role in glucose metabolism by inactivating incretin hormones, which are responsible for stimulating insulin secretion in response to meals and inhibiting glucagon release. By inhibiting DPP-4, sitagliptin enhances the levels of these incretin hormones, leading to increased insulin secretion and reduced blood sugar levels, particularly after meals [20]. Although sitagliptin showed, in silico, the best vina score on the FtsZ catalytic site, it did not demonstrate anti-TB activity. Other studies have observed that sitagliptin shows no direct activity on the avirulent H37Ra strain of Mtb at concentrations of 100 µg/mL [27]. These results coincide with those observed in our study. However, the authors have reported that the co-stimulation of infected macrophages with sitagliptin (100 µg/mL) and isoniazid (0.10 μg/mL) or rifampicin (0.40 μg/mL) promotes antimicrobial activity in a synergetic way [27]. Thus, the use of sitagliptin as an adjuvant in the treatment of TB should be evaluated, as demonstrated previously with metformin [28].

Atovaquone is a potent antimalarial agent that acts as a competitive inhibitor of ubiquinol, specifically targeting the bc1 complex within the mitochondrial electron transport chain. By inhibiting the bc1 complex’s activity, atovaquone disrupts the mitochondrial function of malaria-causing parasites, resulting in decreased ATP production and ultimately hindering their growth and replication [23]. The impairment of mitochondrial function is critical since these parasites rely significantly on aerobic respiration for energy. Our findings indicate that atovaquone lacks antimicrobial activity against H37Rv and MDR TB at concentrations below 0.028 µM. However, some studies suggest that antimicrobial activity may be observed at a concentration of 1 µM [29]. Although the mechanisms of action were not clarified by the authors, the potential use of atovaquone as an anti-TB agent is not dismissed, suggesting its evaluation at higher concentrations. Further investigation is recommended to explore this potential benefit, possibly involving the binding and inhibition of the FtsZ protein.

Paroxetine is a selective serotonin reuptake inhibitor (SSRI) used to treat patients with depression, including fluoxetine and fluvoxamine. Interestingly, we observed that paroxetine decreased mycobacterial growth. This molecule has been demonstrated to have antimicrobial (MIC: 32–152 μg/mL) [21] and antifungal (MIC: 41 μg/mL) [30] effects. However, this is the first report to describe anti-Mtb activity. Although the mechanism of action has not been completely described, the results shown by Cabral et al. propose that paroxetine triggers alterations in membrane integrity and bacterial DNA fragmentation [31]. Furthermore, some authors describe that based on the mechanisms of action of SSRIs, they could act as an inhibitor of bacterial efflux pumps [32]. Thus, toxic metabolic molecules could concentrate inside the bacterial cell; however, this hypothesis needs to be tested. We postulate that the observed antimicrobial effect on Mtb could be attributed, at minimum, to paroxetine’s ability to inhibit FtsZ, as predicted in our in silico model. 

Depression is one of the most common psychological TB comorbidities due to its high prevalence among mental disorders. A recent meta-analysis indicates a high prevalence of comorbid depression (45.19% (95% CI = 38.04–51.37)) among patients with TB [33]. However, depression is not routinely diagnosed or treated as part of TB services. According to WHO-recommended drugs for the treatment of depression in the Mental Health Gap Action Program (mhGAP) guidelines, in patients with comorbid TB, concerns have been raised over potential drug interactions between various SSRIs and isoniazid [34]. At the molecular level, research indicates that isoniazid and SSRIs share similar metabolic pathways. However, current evidence suggests that paroxetine is the safest SSRI to use alongside isoniazid, compared to other SSRIs. Paroxetine is primarily metabolized by CYP2D6, which is only slightly affected by isoniazid, resulting in a minimal risk for drug interactions [35]. Therefore, the use of paroxetine for this comorbidity seems to be safe; however, clinical and retrospective trials need to be conducted to assess the safety of this combination.

Nebivolol is a β-1 adrenergic receptor antagonist that blocks beta-1 receptors, making it a cardio selective β-blocker. This drug also acts on the vascular endothelium by stimulating nitric oxide (NO) synthase, which induces NO-mediated vasodilation. In this study, we have shown for the first time the antimicrobial properties of nebivolol. Moreover, a series of new sulphonamide and carbamate derivatives of the nebivolol drug intermediate exhibited promising antimicrobial and fungal activity [22]. We suggest that structure–activity assays must be evaluated to determine the functionality of nebivolol as an antimicrobial drug.

Our experiments demonstrated that nebivolol, at concentrations of 25 and 10 µg/mL, significantly reduced bacilli levels within macrophages compared to rifampicin treatment, indicating its potential for managing bacterial infections. Nebivolol has been investigated for its potential to modulate the immune response, including its effects on cytokines and chemokines involved in the immune response against bacterial infections. Studies have shown that nebivolol can influence immune pathways beyond its primary role as a beta-1 adrenergic receptor antagonist. Nebivolol’s promotion of nitric oxide (NO) production may contribute to a more balanced immune response during bacterial infections, potentially aiding in the control of microbial growth and reducing tissue damage caused by excessive inflammation [36].

Moreover, nebivolol has been shown to decrease levels of pro-inflammatory cytokines such as tumor necrosis factor-alpha (TNF-α) and interleukin-6 (IL-6) in various inflammatory conditions [37]. These cytokines are key mediators of the inflammatory response and are implicated in the pathogenesis of infectious diseases, including TB. By reducing their levels, nebivolol may help mitigate the inflammatory cascade associated with bacterial infections, thereby supporting the host’s immune defense mechanisms.

Further research is warranted to elucidate the precise mechanisms by which nebivolol modulates immune responses during bacterial infections, particularly those caused by Mtb. Understanding these mechanisms could pave the way for novel therapeutic strategies that harness nebivolol’s immunomodulatory properties alongside conventional antimicrobial treatments. In conclusion, nebivolol’s ability to modulate immune responses, including its effects on cytokines and NO production, suggests potential applications in adjunctive therapies for bacterial infections.

Our results demonstrate that our FtsZ inhibitor candidates exhibit significant mycobactericidal activity. However, further studies are necessary to determine whether these candidates bind to FtsZ in vitro or to any other essential Mtb molecules, as well as to assess their intracellular activity.

## 5. Conclusions

In summary, the selection of FtsZ inhibitors provides a novel strategy to control Mtb. Thus, in addition to its pharmacological effects, we have described a previously unknown anti-Mtb drug. Further investigations are necessary to clarify the functionality of such molecules as anti-Mtb drugs in clinical trials.

## Figures and Tables

**Figure 1 microorganisms-12-01505-f001:**
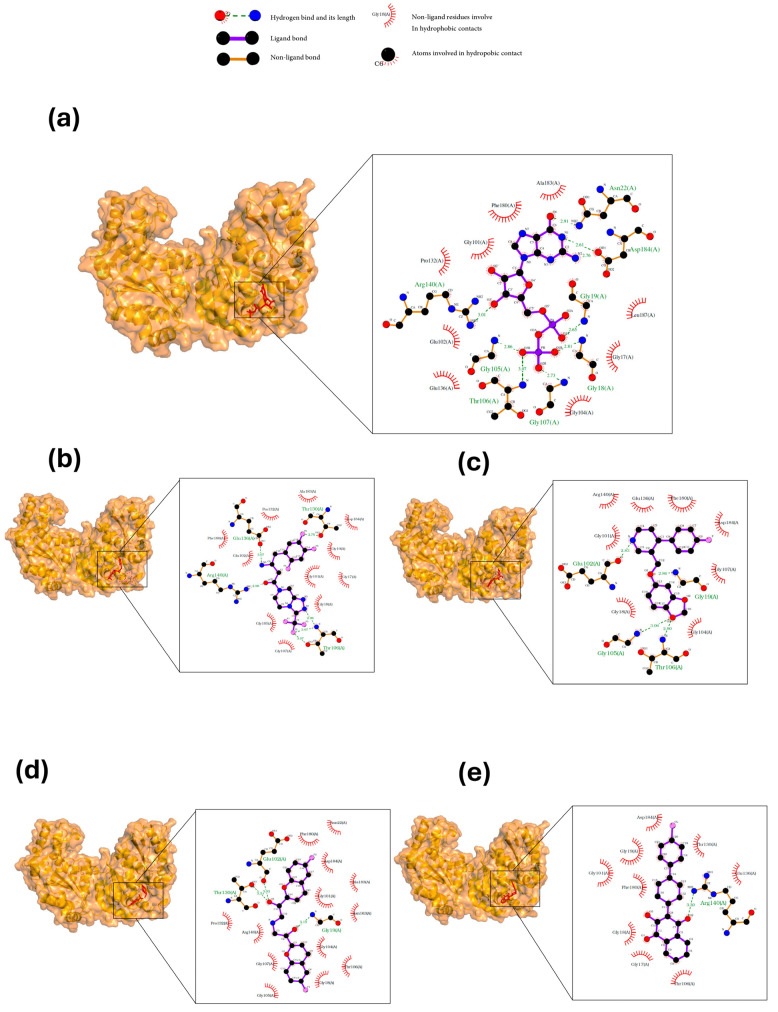
A three-dimensional model of the interactions between crystallized ligands with (**a**) GDP, (**b**) sitagliptin, (**c**) paroxetine, (**d**) nebivolol, and (**e**) atovaquone with the FtsZ protein. (**b**–**e**) The inset represents the interaction of each drug with the 9 amino acids of the FtsZ Mtb protein.

**Table 1 microorganisms-12-01505-t001:** Vina scores from FDA-approved drugs near to −8.5 kCal/mol from control ligand (GDP).

ZINC ID	Vina Score (kCal/mol)	FDA- Approved Drug	Structure	Reported Activity	Amino Acid Residues	Citation
1489478	−9.4	Sitagliptin	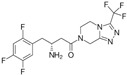	Inhibition of dipeptidyl peptidase-4 (DPP-4)	GLU-136 ARG-140	[20]
527386	−9	Paroxetine	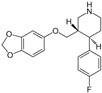	Selective serotonin reuptake inhibitor (SSRI)	THR-106 GLY-105 GLY-19 GLU-102	[21]
4213946	−8.6	Nebivolol	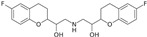	Beta-blocker	ASN-22 THR-106 GLY-19 ARG-140 GLU-136	[22]
12504271	−8.5	Atovaquone	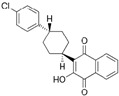	Antimicrobial, antifungal	ARG-140	[23]
	−8.5	GDP guanosine diphosphate	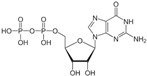		ASP-184 GLU-136 ARG-140 GLY-19 GLY-18 ANS-22 THR-106 GLY-105 GLY-107	

**Table 2 microorganisms-12-01505-t002:** Determination of Minimal Inhibitory Concentrations (MICs) of FtsZ inhibitor candidate molecules against H37Rv and MDR strains.

FDA-Approved Drug	H37Rv (µg/mL)	MDR (µg/mL)
Rifampicin	0.5	>1
Streptomycin	0.5	0.5
Sitagliptin	>150	>150
Atovaquone	>10	>10
Paroxetine	25	25
Nebivolol	25	25

## Data Availability

The data that support the findings of this study are available from the corresponding author, [ARC and JELR], upon reasonable request.

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
