# Peer review of "The Repurposing of FDA-Approved Drugs as FtsZ Inhibitors against Mycobacterium tuberculosis: An In Silico and In Vitro Study"

_microorganisms, 2024, doi:10.3390/microorganisms12081505_

Round 1

Reviewer 1 Report

Comments and Suggestions for Authors

This work addresses the consideration of the possibility of using several FDA-approved antibiotic drugs, originally not intended for the treatment of tuberculosis, as antituberculosis drugs. Taking into account the extremely long time required for approving and introducing newly developed substances into clinical practice, such an alternative is of high interest. The study looks carefully carried out and combines both practical tests and molecular structure-based analysis, which supports the observations from a biophysical point of view.

Thus, I recommend the acceptance of this manuscript after minor revisions aimed at making more clear some details of the procedure description:

  1. Respectively the Alamar Blue-based tests: it should be stated, after which time of growth after the addition of indicator medium, the colourimetric measurements have been done, 24 hours, as it follows from the standard protocol? Which method was applied to determine the MIC, visual, fluorometric, or quantitative colourimetric? Details and criteria should be provided.
  2. Optionally, it is advisable to provide photos (and, maybe additionally, fluorescence intensity, if it was applied) of the plates used to determine MIC in as a supplementary material.
  3. Respectively mycobacterial growth curves (Fig. 2): the details of the procedure should be given in the section Materials and Methods; the plots miss the unit under the abscissa; is it in days?

Author Response

We would like to acknowledge the reviewers for their comments and suggestions. We took all the comments and the suggestions of the Editor into account. More precisely:

  1. Respectively the Alamar Blue-based tests: it should be stated, after which time of growth after the addition of indicator medium, the colourimetric measurements have been done, 24 hours, as it follows from the standard protocol? Which method was applied to determine the MIC, visual, fluorometric, or quantitative colourimetric? Details and criteria should be provided.

This information has been added from line 115 to 198 in the manuscript.

  1. Optionally, it is advisable to provide photos (and, maybe additionally, fluorescence intensity, if it was applied) of the plates used to determine MIC in as a supplementary material. 

We greatly appreciate your feedback, and we would like to include representative images for each of the essays. However, we have decided not to present them because the quality of the existing photos does not meet the standards required by the journal. Additionally, there is not enough time to take new photographs. We hope for your understanding in this matter.

  1. Respectively mycobacterial growth curves (Fig. 2): the details of the procedure should be given in the section Materials and Methods; the plots miss the unit under the abscissa; is it in days?

This information has been updated in the figure 2.

Reviewer 2 Report

Comments and Suggestions for Authors

The introduction of the present manuscript entitled “Repurposing of FDA-approved drugs as FtsZ inhibitor against Mycobacterium tuberculosis: An in silico and in vitro study” by Tovar-Nieto et al sets the stage for the study, providing comprehensive background information on the global impact of tuberculosis (TB) and the need for new therapeutic approaches due to the rise of multidrug-resistant (MDR) strains. The authors have explained well why targeting the FtsZ protein is a good strategy for TB treatment, citing relevant studies to show the importance of their research. The introduction is well-structured and informative, providing a good foundation for the study.

Additionally, the methodology section is detailed and describes the various in silico and in vitro techniques used to identify and evaluate FDA-approved drugs as potential FtsZ inhibitors. The molecular docking procedures, cell culture conditions, and assays (Microplate Alamar Blue Assay, cytotoxicity assay, and UFCs/mL assay) are thoroughly explained.

Moreover, the results are presented in a clear and logical manner, with the utilization of tables and figures to illustrate their findings. The identification of paroxetine and nebivolol as promising FtsZ inhibitors, their demonstrated antimycobacterial activity, and the in vitro validation of their efficacy against Mtb and MDR strains are well-documented. The statistical analyses are appropriately conducted, and the significance of the results is clearly stated.

Finally, the authors provide a balanced view of the strengths and limitations of their study, acknowledging the need for further research to fully understand the mechanisms of action and to evaluate the clinical potential of the identified drugs. The conclusion summarizes the main findings and emphasizes the novelty and significance of the research.

Suggestions/corrections to improve the manuscript:

1-   Line 303: In summary (please correct the word, thanks).

2-   Figures 1: it should be necessary to make a zoom in the region of interest because it is not possible to see the position of the ligands appropriated.

3-   Providing comparative data on the efficacy of paroxetine and nebivolol relative to existing TB treatments (e.g., first-line drugs like isoniazid and rifampicin) would offer a clearer perspective on the potential advantages and limitations of these repurposed drugs.

4-   While the limitations of the study are acknowledged, a more detailed discussion of specific challenges (e.g., potential drug resistance, side effects, pharmacokinetics) and proposed future research directions would provide valuable context for readers and researchers interested in building upon this work.

Author Response

Microorganisms-3105922

Repurposing of FDA-approved drugs as FtsZ inhibitor against Mycobacterium tuberculosis: An in silico and in vitro study.

We would like to acknowledge the reviewers for their comments and suggestions. We took all the comments and the suggestions of the Editor into account. More precisely:

Suggestions/corrections to improve the manuscript:

  • Line 303: In summary (please correct the word, thanks).

 We’re agreed. It has been corrected (line 405)

  • Figures 1: it should be necessary to make a zoom in the region of interest because it is not possible to see the position of the ligands appropriated.

We’re agreed. It has been corrected in the figure 1

  • Providing comparative data on the efficacy of paroxetine and nebivolol relative to existing TB treatments (e.g., first-line drugs like isoniazid and rifampicin) would offer a clearer perspective on the potential advantages and limitations of these repurposed drugs.

During our study, we included growth controls along with the use of first-line drugs. Consequently, we have included in the discussion the essential comparisons with first-line drugs that were evaluated in all trials.

  • While the limitations of the study are acknowledged, a more detailed discussion of specific challenges (e.g., potential drug resistance, side effects, pharmacokinetics) and proposed future research directions would provide valuable context for readers and researchers interested in building upon this work.

This information has been added to the manuscript in a discussion section.

Reviewer 3 Report

Comments and Suggestions for Authors

Due to my limited experts, the computational part is thus only reviewed. Four drugs with favorable Gibbs free energy scores were identified from 1947 FDA-approved drugs in the Zinc20 database with Autodock Vina. The activity of these drugs against mycobacterium tuberculosis were then tested by several in vitro and in vivo experiments and they show positive results. Thus, the manuscript is recommended for acceptance expect the following issues:

(1) In Figure 1, the interactions for screened protein-ligand complexes were hardly visible. A detailed 3D interaction plots in the binding pocket at the atomistic level should be provided.

(2) In line 91 on page2, “-8.5kCal” should be “-8.5 kcal/mol”.

Author Response

Microorganisms-3105922

Repurposing of FDA-approved drugs as FtsZ inhibitor against Mycobacterium tuberculosis: An in silico and in vitro study.

We would like to acknowledge the reviewers for their comments and suggestions. We took all the comments and the suggestions of the Editor into account. More precisely:

issues:

  1. In Figure 1, the interactions for screened protein-ligand complexes were hardly visible. A detailed 3D interaction plots in the binding pocket at the atomistic level should be provided.

We’re agreed. It has been corrected in a figure 1 .

  1. In line 91 on page2, “-8.5kCal” should be “-8.5 kcal/mol”.

We’re agreed. It has been corrected (line 96)